# Sensitivity to vocal emotions emerges in newborns at 37 weeks gestational age

**Xinlin Hou[1†], Peng Zhang[1,2†], Licheng Mo[3], Cheng Peng[1], Dandan Zhang[3,4,5,6]\***

[1]Department of Pediatrics, Peking University First Hospital, Beijing, China; [2]Department of Pediatrics, Miyun Country Maternal and Child Health Hospital, Beijing, China; [3]Institute of Brain and Psychological Sciences, Sichuan Normal University, Chengdu, China; [4]Shenzhen-Hong Kong Institute of Brain Science, Shenzhen, China; [5]China Center for Behavioral Economics and Finance, Southwestern University of Finance and Economics, Chengdu, China; [6]School of Psychology, Chengdu Medical College, Chengdu, China

**\*For correspondence:**
zhangdd05@gmail.com

[†]These authors contributed equally to this work

**Competing interest:** The authors declare that no competing interests exist.

**Abstract** Emotional responsiveness in neonates, particularly their ability to discern vocal emotions, plays an evolutionarily adaptive role in human communication and adaptive behaviors. The developmental trajectory of emotional sensitivity in neonates is crucial for understanding the foundations of early social-emotional functioning. However, the precise onset of this sensitivity and its relationship with gestational age (GA) remain subjects of investigation. In a study involving 120 healthy neonates categorized into six groups based on their GA (ranging from 35 and 40 weeks), we explored their emotional responses to vocal stimuli. These stimuli encompassed disyllables with happy and neutral prosodies, alongside acoustically matched nonvocal control sounds. The assessments occurred during natural sleep states using the odd-ball paradigm and event-related potentials. The results reveal a distinct developmental change at 37 weeks GA, marking the point at which neonates exhibit heightened perceptual acuity for emotional vocal expressions. This newfound ability is substantiated by the presence of the mismatch response, akin to an initial form of adult mismatch negativity, elicited in response to positive emotional vocal prosody. Notably, this perceptual shift's specificity becomes evident when no such discrimination is observed in acoustically matched control sounds. Neonates born before 37 weeks GA do not display this level of discrimination ability. This developmental change has important implications for our understanding of early social-emotional development, highlighting the role of gestational age in shaping early perceptual abilities. Moreover, while these findings introduce the potential for a valuable screening tool for conditions like autism, characterized by atypical social-emotional functions, it is important to note that the current data are not yet robust enough to fully support this application. This study makes a substantial contribution to the broader field of developmental neuroscience and holds promise for future research on early intervention in neurodevelopmental disorders.

## eLife assessment

This is an **important** study on changes in newborns' neural abilities to distinguish auditory signals at 37 weeks of gestation. The evidence of change in neural discrimination as a function of gestational age is **convincing**, but, as the authors acknowledge, further control of the acoustic signals and infants' language environment is necessary for the results to be used in clinical applications. The work contributes to the field of neurodevelopment.

## Introduction

Emotions represent a fundamental aspect of human social interaction, serving as a compelling subject of inquiry within the disciplines of neuroscience, psychology, and psychiatry. Over the course of evolution, the human brain has evolved to possess a heightened sensitivity to the emotional expressions of others (*Lindquist et al., 2012*). Remarkably, even prior to the full maturation of their visual system, human infants exhibit a remarkable ability to discern vocal emotions (*Blasi et al., 2011*; *Soderstrom et al., 2017*; *Vaish and Striano, 2004*). Prosodic elements of speech, including pitch, intensity, and rhythm, function as universal and non-linguistic channels for emotional communication (*Latinus and Belin, 2011*). Numerous studies have established that infants, including those who have not yet acquired language, exhibit differentiated responses to emotional prosody conveying happiness, fear, anger, and sadness within the age range of 2–12 months (e.g. *Caron et al., 1988*; *Fernald, 1993*; *Graham et al., 2013*; *Grossmann et al., 2010*; *Singh et al., 2002*; *Walker-Andrews and Grolnick, 1983*; *Zhao et al., 2021*).

More specifically, during the very early stages of postnatal life, often termed the neonatal period (encompassing infants under four weeks of age), compelling evidence points to the presence of emotion-specific responses to emotional cues conveyed through vocal prosody. These responses have been identified through various measurement methods, including assessments of eye-opening scores (*Mastropieri and Turkewitz, 1999*), event-related potentials (*Cheng et al., 2012*), and near-infrared spectroscopy (*Zhang et al., 2019*). However, prior research has primarily focused on the perception and discrimination of emotions among traditionally defined term neonates, a group that includes infants born within a five-week span (37–41 weeks) of GA, treating them as a homogenous cohort. This raises a crucial question: when does emotional sensitivity begin to manifest in newborns? Does it exist in preterm neonates (GA <37 weeks)? And does it vary among neonates born at early term (GA = 37–38 weeks) and full-term (GA = 39–40 weeks), as defined by the refined 'term' classification (*Spong, 2013*)? Surprisingly, to date, no study has explored emotion processing in neonates with varying GAs. The discovery of this developmental milestone not only advances our understanding of the cognitive mechanisms underlying human social-emotional functioning but also provides valuable insights for early diagnosis of neurodevelopmental disorders, such as autism (*Jones et al., 2014*; *Molnar-Szakacs et al., 2021*).

The principal objective of this study is to investigate emotional responses in neonates across a range of GAs, spanning from 35 to 40 weeks, and to determine whether their heightened sensitivity to emotional voices is influenced by GA. To achieve this, we utilized the oddball paradigm in conjunction with an event-related potential (ERP) component known as mismatch negativity (MMN) to probe the neurobiological encoding of emotional voices in the neonatal brain. MMN is an auditory ERP component that demonstrates a negative shift in response to deviant sounds when compared to standard sounds (*Näätänen et al., 2007*). Importantly, it can be elicited without requiring the subject's attention, making it particularly suitable for recording in young infants (*Cheour et al., 1998*; *Cheour et al., 2002*). It is worth noting that in neonates, this ERP component often manifests as a positive response rather than the traditional MMN (e.g. *Cheng et al., 2012*; *Chládková et al., 2021*; *Kostilainen et al., 2020*; *Richard et al., 2022*; *Thiede et al., 2019*; *Virtala et al., 2022*; *Winkler et al., 2003*; *Winkler et al., 2009*), leading many researchers to refer to it as the mismatch response (MMR) in the neonatal brain.

In our study, we exposed neonates to speech samples characterized by positive (i.e. happy) and neutral prosodies. Our selection of positive emotions over negative ones (e.g. fear, sadness, or anger) was guided not only by ethical considerations but also by previous research indicating an early preference for positive emotions in neonates (*Farroni et al., 2007*; *Mastropieri and Turkewitz, 1999*; *Zhang et al., 2019*; with the exception of *Cheng et al., 2012*). Additionally, to eliminate the possibility of neonates distinguishing emotional voices solely based on their low-level acoustic features, we included another set of control sounds. These nonvocal stimuli were meticulously matched with their vocal prosodic counterparts in terms of mean intensity and fundamental frequency (*Cheng et al., 2012*). Consequently, our primary objective is to pinpoint the developmental stage (i.e. the GA group) at which the discrimination between happy and neutral stimuli becomes apparent for emotional voices while remaining absent for acoustically matched control sounds.

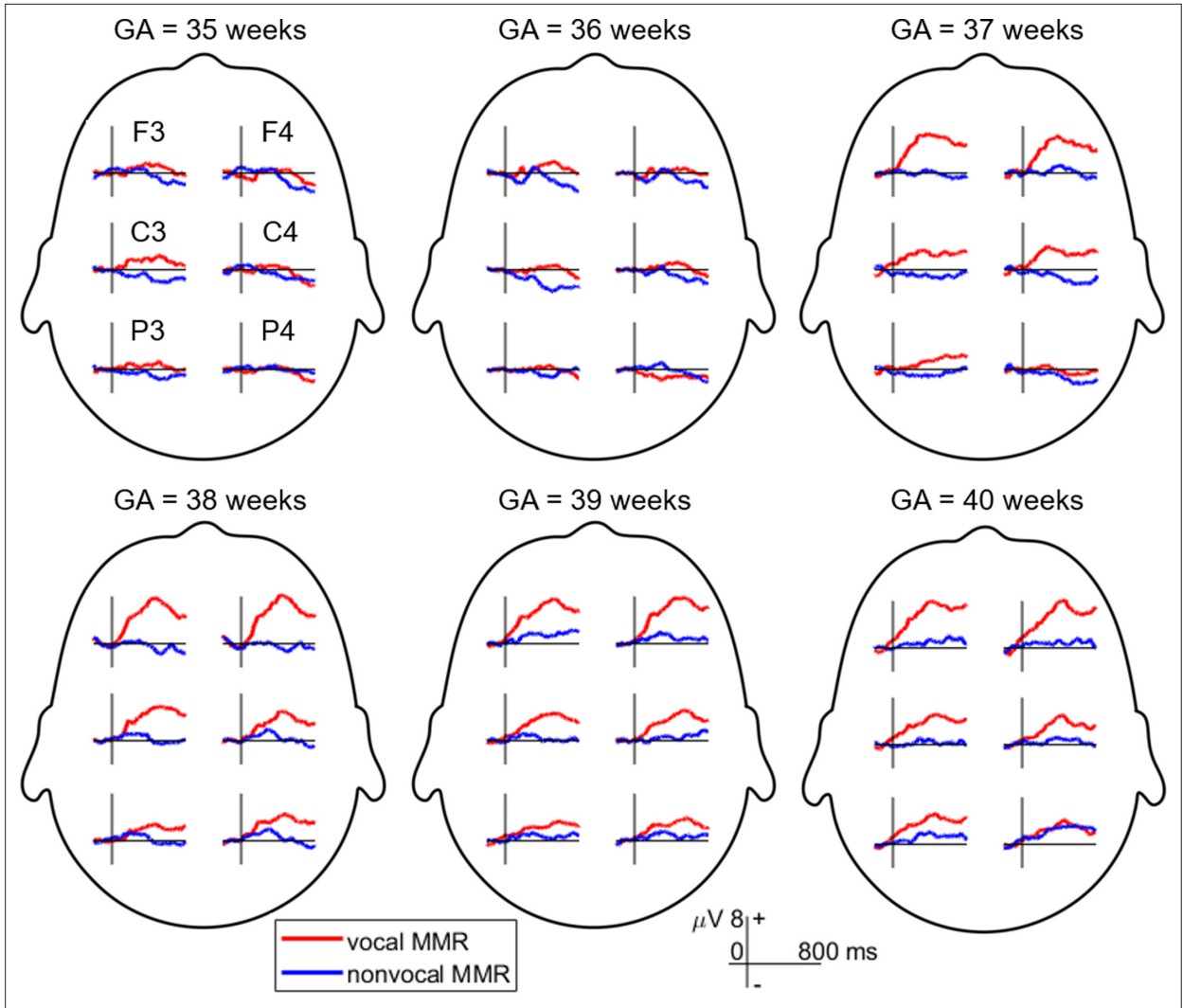

**Figure 1.** Mismatch response (MMR) waveforms were recorded across six electrodes in different gestational age (GA) groups. The MMR is extracted by subtracting the event-related potential (ERP) elicited by the standard stimulus (neutral condition) from the ERP evoked by the deviant stimulus (happy condition).

## Results

The MMR was extracted using ERP difference waves, computed by subtracting the ERP evoked by the standard stimulus (neutral sound) from the ERP evoked by the deviant stimulus (happy sound) (*Näätänen et al., 2007*). Brain electrical activity was recorded from the F3, F4, C3, C4, P3, and P4 sites following the international 10/20 system. However, this study primarily focused on data from electrodes F3 and F4, as the neonatal MMR exhibits a frontal distribution (*Cheng et al., 2012*; *Cheour et al., 2002*). *Figure 1* displays MMR waveforms recorded from all six electrodes.

Initially, we conducted a three-way repeated measures ANOVA on the mean MMR amplitudes (time window: 150–400 ms after sound onset) with factors including condition (vocal/nonvocal), hemisphere (left/right frontal, i.e. F3/F4) as within-subjects factors, and neonatal group (GA = 35, 36, 37, 38, 39, and 40 weeks) as the between-subjects factor. However, neither the main effect nor the interaction effects involving the hemisphere factor were statistically significant. In particular, the main effect of the hemisphere is not significant, $F(1,114) = 0.153$, p=0.696, $\eta_p^2 = 0.001$. The interaction between hemisphere and group is not significant, $F(5,114) = 0.249$, p=0.940, $\eta_p^2 = 0.011$. The interaction between of hemisphere and stimuli is not significant, $F(1,114) = 0.474$, p=0.492, $\eta_p^2 = 0.004$. The three-way interaction is not significant, $F(5,114) = 0.666$, p=0.650, $\eta_p^2 = 0.028$. Consequently, we removed the hemisphere factor and averaged the MMR waveforms recorded at the F3 and F4 electrodes.

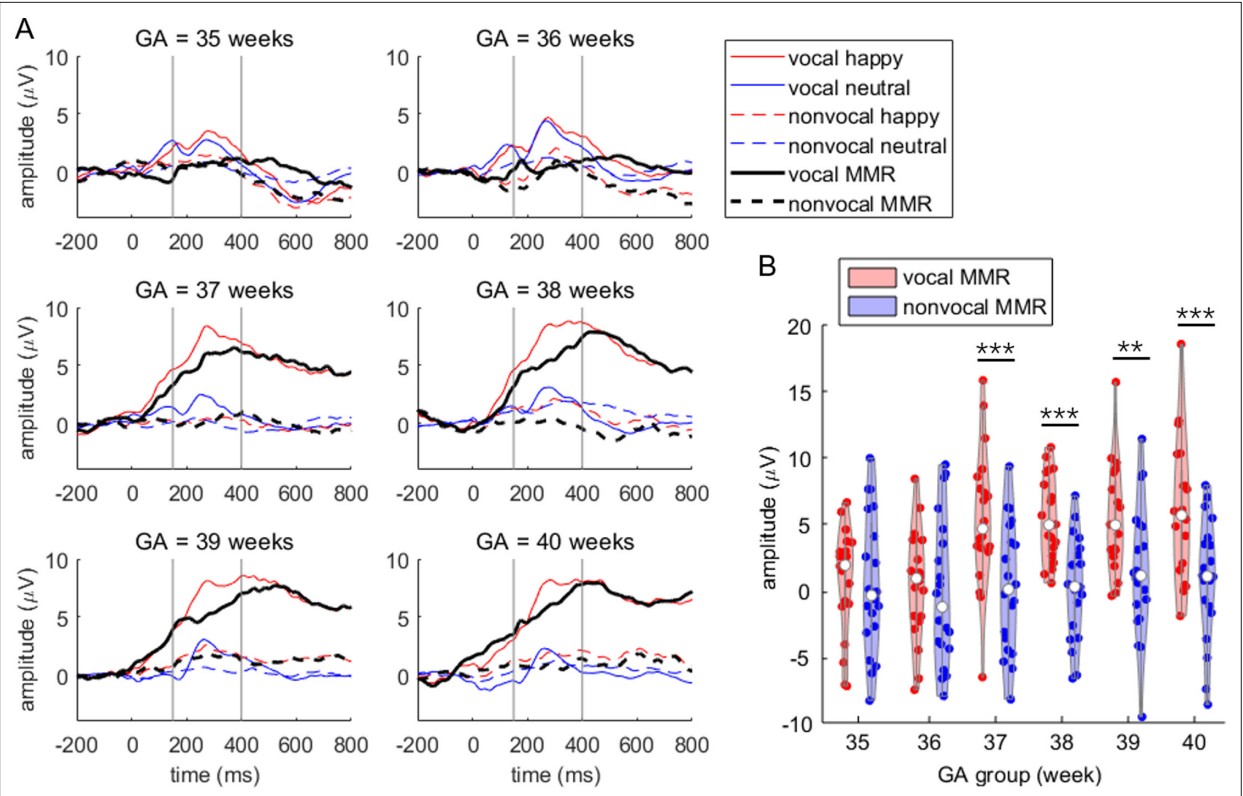

**Figure 2.** Primary mismatch response (MMR) results. (**A**) Event-related potential (ERP) waveforms (averaged at the F3 and F4 electrodes) across the six gestational age (GA) groups. The time window for assessing MMR amplitude is indicated between the two vertical gray lines. (**B**) Violin plot illustrating MMR amplitudes in the six GA groups. Simple effect analysis was performed with pairwise comparisons, corrected using the Bonferroni method: **p<0.010, ***p<0.001.

Subsequently, we performed a two-way repeated measures ANOVA with condition and group as the two factors. The main effect of stimuli was significant, $F(1,114) = 38.827$, $p<0.001$, $\eta_p^2 = 0.254$. Specifically, vocal stimuli elicited larger MMRs (mean ± standard deviation: 3.839 ± 4.855 µV) compared to nonvocal stimuli (0.496 ± 4.779 µV). The main effect of group was also significant, $F(5,114) = 3.228$, $p=0.009$, $\eta_p^2 = 0.124$. In general, MMR amplitudes were smaller in the GA35 (0.590 ± 4.579 µV) and GA36 (0.141 ± 4.807 µV) groups compared to the GA37 (2.801 ± 5.585 µV), GA38 (2.760 ± 4.382 µV), GA39 (3.401 ± 4.871 µV), and GA40 groups (3.311 ± 5.491 µV). However, no significant differences were found in pairwise comparisons after Bonferroni adjustment for multiple comparisons.

The interaction between stimuli and group was significant, $F(5,114) = 3.127$, $p=0.011$, $\eta_p^2 = 0.121$ (as shown in *Figure 2*). Simple effect analysis revealed that MMR amplitudes were larger in the vocal condition compared to the nonvocal condition in the GA37 ($F(1,114) = 15.254$, $p<0.001$, $\eta_p^2 = 0.118$; vocal = 5.367 ± 5.165 µV, nonvocal = 0.235 ± 4.847 µV), GA38 ($F(1,114) = 16.072$, $p<0.001$, $\eta_p^2 = 0.124$; vocal = 5.394 ± 3.145 µV, nonvocal = 0.126 ± 3.861 µV), GA39 ($F(1,114) = 8.393$, $p=0.005$, $\eta_p^2 = 0.069$; vocal = 5.305 ± 4.011 µV, nonvocal = 1.498 ± 4.998 µV), and GA40 groups ($F(1,114) = 14.482$, $p<0.001$, $\eta_p^2 = 0.113$; vocal = 5.811 ± 5.298 µV, nonvocal = 0.811 ± 4.546 µV). However, there were no significant differences in MMR amplitudes between the two kinds of stimuli in the GA35 ($F(1,114) = 0.026$, $p=0.873$, $\eta_p^2 < 0.001$; vocal = 0.695 ± 4.031 µV, nonvocal = 0.485 ± 5.173 µV) and GA36 groups ($F(1,114) = 0.236$, $p=0.628$, $\eta_p^2 = 0.002$; vocal = 0.460 ± 4.104 µV, nonvocal = –0.179 ± 5.511 µV).

Further analysis revealed that vocal stimuli evoked varying MMR amplitudes across groups, $F(5,114) = 6.768$, $p<0.001$, $\eta_p^2 = 0.229$. Specifically, the MMRs evoked by vocal stimuli were smaller in the GA35 group compared to GA37 ($p=0.014$), GA38 ($p=0.013$), GA39 ($p=0.017$), and GA40 groups ($p=0.005$). Similarly, the MMRs evoked by vocal stimuli were smaller in the GA36 group compared to GA37 ($p=0.008$), GA38 ($p=0.008$), GA39 ($p=0.009$), and GA40 groups ($p=0.003$). However, nonvocal stimuli did not elicit significantly different MMR amplitudes across groups, $F(5,114) = 0.300$, $p=0.912$, $\eta_p^2 = 0.013$.

## Discussion

The current study elucidates a pivotal developmental change in neonatal emotional responsiveness by investigating their ability to perceive vocal emotions. The findings illuminate a distinct turning point at 37 weeks of GA, representing the onset of heightened perceptual acuity for emotional vocal expressions. This change is particularly evident in the robust MMR to positive emotional vocal prosody. Significantly, the absence of this discrimination ability when acoustically matched control sounds were presented underscores the specificity of this developmental shift towards emotional voice processing. Our identification of the 37 week GA mark aligns with previous research, which indicated emotional sensitivity in term neonates born at or after 37 weeks of gestation (*Cheng et al., 2012*; *Farroni et al., 2007*; *Mastropieri and Turkewitz, 1999*; *Zhang et al., 2019*) and in preterm neonates (GA <37 weeks) tested at term age (*Kostilainen et al., 2020*). Notably, our findings also reveal that neonates born before 37 weeks GA do not exhibit these emotional discrimination abilities.

In the final trimester of pregnancy, the human brain undergoes a period of rapid and continuous changes in neural structure and cognitive functions (*Bayer et al., 1993*; *Clancy et al., 2007*). Although there is no direct evidence to support the notion that preterm neonates can decode vocal emotions, they have displayed an aptitude for processing speech stimuli in social contexts. For example, neonates born at or after 29 weeks GA have shown a preference for infant-directed speech, characterized by a high pitch, exaggerated pitch modifications, and a slow rate (*Butler et al., 2014*). This preference has been associated with increased visual attention, heightened alertness (*Eckerman et al., 1994*), reduced heart rate (*White-Traut et al., 1997*), and enhanced speech differentiation in premature babies (*Richard et al., 2022*). Additionally, neonates born at or after 30 weeks GA have demonstrated an age-related increase in sensitivity to maternal voices (*D Chorna et al., 2018*; *Key et al., 2012*), leading to beneficial effects on cognitive and neurobehavioral development (*Caskey et al., 2011*; *Picciolini et al., 2014*; see *Provenzi et al., 2018* for a review). These effects encompass improved feeding behaviors, heightened responsiveness (*Katz, 1971*; *Krueger et al., 2010*), enhanced weight gain (*Zimmerman et al., 2013*), and activated auditory cortical plasticity (*Webb et al., 2015*). Both infant-directed speech and maternal voices feature extensive pitch modulation, and the preference for these emotionally prosodic-like voices during the preterm stage may prepare the developing brain to discriminate vocal emotions at 37 weeks GA, as demonstrated in this study.

Traditionally, 37 weeks of gestation served as the benchmark for fetal maturity, and term infants born within the 37–41 weeks GA range were generally considered healthy, forming a homogenous group. Recent insights, however, have unveiled variations in physical and cognitive maturation within this 5 week span of full-term pregnancy. Research indicates that neonates born at 37–38 weeks GA face increased risks of neonatal mortality and pediatric respiratory, neurologic, and endocrine morbidities compared to those born at 39–41 weeks GA (*Cahen-Peretz et al., 2022*; *Clark et al., 2009*; *Edwards et al., 2013*; *Ghartey et al., 2012*; *Paz Levy et al., 2017*; *Sengupta et al., 2013*; *Tita et al., 2009*). Furthermore, a dose-response relationship inversely linking GA to the risk of developmental delay has been identified in infants from preterm to full-term births (*Rose et al., 2013*; *Schonhaut et al., 2015*). Early birth (34–38 weeks GA) has also been found to have a detrimental impact on child development and academic achievement during school age (*Bentley et al., 2016*; *Chan et al., 2016*; *Dong et al., 2012*; *Hedges et al., 2021*; *Murray et al., 2017*; *Nielsen et al., 2019*; *Noble et al., 2012*). Consequently, the definition of a full-term pregnancy has been narrowed to a two-week window starting at 39 weeks (*Spong, 2013*), with nonmedically indicated deliveries between 37 and 38 weeks of gestation discouraged (*ACOG Committee Opinion, 2019*). While accumulating evidence underscores the adverse effects of the traditional 37 week threshold, our findings contribute to the limited body of research suggesting that neonatal social-emotional functioning may reach a development milestone at 37 weeks GA.

When interpretating the current findings, it is important to consider that the nonvocal control sounds utilized in this study may not have adequately eliminated all low-level acoustic properties that could aid neonatal discrimination. Specifically, while the nonvocal control counterparts retained the fundamental frequency (f0) of the emotional prosodic voices, they did not replicate the burst of energy associated with consonants. Consequently, it cannot be ruled out that neonates utilized consonant characteristics to discriminate emotional prosodies conveyed by disyllables. Additionally, the nonvocal sounds were generated using a simple filtering method, resulting in certain vocal-like

components persisting in these control sounds. The limitations of the control sound materials should be given greater consideration in future replication or further research.

Furthermore, there is a compelling need for future investigations to expand upon the present findings by incorporating a broader array of emotional stimuli. Non-speech emotional vocalizations, such as laughter, crying, or retching, as well as natural emotional auditory cues like thunder, flowing water, hissing snakes, and bird calls, offer a rich spectrum of emotional materials that have been shown to engage the perceptual faculties of neonates and infants (*Blasi et al., 2011*; *Erlich et al., 2013*). This multifaceted approach could illuminate whether the developmental milestone observed at 37 weeks GA is specific to the processing of emotional prosodic speech and vocal expressions, or if it extends to encompass a broader range of both artificial and natural emotional auditory cues. Moreover, the use of non-speech emotional stimuli aids in resolving the debate between nature and nurture concerning the onset of emotional sensitivity at 37 weeks GA. It cannot solely attribute the current finding of discrimination to the innate maturational explanation, given that the auditory system becomes functional at the end of the second trimester of pregnancy, allowing exposure to spoken language in utero to influence the development of speech perception (*DeCasper and Spence, 1986*; *Moon et al., 2013*; *Partanen et al., 2013*). The ability to discriminate prosodic emotions starting at 37 weeks GA could stem from additional exposure in utero to speech. Future exploration is needed to definitively investigate prenatal learning by utilizing emotional sounds that are infrequently encountered in the prenatal environment. Finally, the inclusion of non-speech emotional materials may offer insights into the potential right lateralization of emotional processing in the neonatal brain. While prior studies (including some cited herein) have identified right lateralization for emotional processing in full-term neonates (*Cheng et al., 2012*; *Zhang et al., 2019*; see *Bisiacchi and Cainelli, 2022* for a comprehensive review), the introduction of non-speech materials can help disentangle the confounding effects of left lateralization, which is associated with language processing and has been identified in both preterm (*Mahmoudzadeh et al., 2013*) and full-term neonates (*Kotilahti et al., 2010*; *May et al., 2018*; *Peña et al., 2003*; *Sato et al., 2012*; *Vannasing et al., 2016*; *Wu et al., 2022*).

A more comprehensive understanding of the developmental trajectory of emotional sensitivity has the potential to revolutionize decision-making in the final weeks of pregnancy and the identification of newborns at risk of emotional and neurodevelopmental disorders, particularly autism. Individuals with autism often exhibit atypical perceptual and neural processing of emotional information, including emotional prosodic voices (*Kuhl et al., 2005*; *Lindström et al., 2018*; *Van Lancker et al., 1989*; *Wang et al., 2007*; for comprehensive reviews, see *Frühholz and Staib, 2017*; *Yeung, 2022*). While previous studies have indicated that social-emotional behavioral indicators typically begin to demonstrate predictive power for autism from the second year of life (*Gliga et al., 2014*; *Jones et al., 2014*), brain functional indicators of emotional processing during infancy, especially within the first year of life, have already shown their predictive value (*Ayoub et al., 2022*; *Clairmont et al., 2021*; *Molnar-Szakacs et al., 2021*). For instance, infants subsequently diagnosed with autism displayed a smaller amplitude and shorter duration of the negative central (Nc) component at six months of age when viewing smiling faces compared to toys, a pattern not observed in infants who were subsequently undiagnosed (*Jones et al., 2016*). Additionally, while the Nc and P400 components were able to distinguish between smiling, fearful, and neutral facial expressions in typically developing 9-to-10-month-old infants, these EEG indicators failed to differentiate emotional faces in infants at high risk for autism (*Di Lorenzo et al., 2021*; *Key et al., 2015*). Moreover, it has been observed that infants at high risk for autism exhibit diminished activation in the fusiform gyrus and hippocampus compared to healthy controls when exposed to sad cries between the ages of 4 and 7 months (*Blasi et al., 2015*). The fusiform gyrus, a region crucial for face perception and memory, and the hippocampus, which plays a significant role in general learning and memory processes, are both implicated in this phenomenon (*Lisman et al., 2017*; *Rossion et al., 2024*). Consequently, the hippocampus-fusiform network, essential for the development of social cognitive skills, may serve as a predictive indicator for the onset of autism. Building upon these existing studies, our research suggests that the neonatal MMR in response to emotional voices could potentially serve as an early screening indicator for autism. However, we acknowledge that the current data are not yet robust enough to fully support this recommendation. We advocate for future longitudinal studies with more rigorous experimental materials and designs to further explore the predictive role of this neurophysiological indicator, which could ultimately facilitate early diagnosis and intervention for social-emotional disorders.

In summary, this study highlights a pivotal developmental change – the emergence of heightened perceptual acuity for emotional vocal expressions at 37 weeks GA. It is important to note that neonates' perceptual sensitivity at this stage is unlikely to be associated with a deep conceptual understanding of emotions. Nevertheless, this unique discrimination ability in early life may serve as a foundational building block for the later development of emotional and social cognition. Overall, this work deepens our understanding of neonatal social and emotional development and suggests a potential avenue for supporting early diagnosis of neurodevelopmental disorders, where early detection is critical for effective intervention.

## Materials and methods
### Subjects
The research received approval from both the Ethical Committee of Peking University First Hospital and the Chinese Clinical Trial Registry (ChiCTR2300069898). Initially, we planned to include 120 healthy neonates, with 60 being boys, in the data analysis. These participants were categorized into six groups based on their GA, specifically 35, 36, 37, 38, 39, and 40 weeks, with each group comprising twenty subjects. For instance, the GA35 group comprised neonates with GA ranging from 35 weeks plus 0 day to 6 days. However, we ultimately recruited 198 neonates to obtain 120 valid datasets due to the non-cooperation of newborns (n=75) or technical issues (n=3). Specially, 11, 12, 11, 14, 13, and 14 neonates were excluded from data analysis in the GA35, GA36, GA37, GA38, GA39, and GA40 groups, respectively, due to crying or irritable movements during EEG device preparation and EEG recording.

The mothers of these neonates were monolingual and nurtured their babies in a native language environment. All neonates participated in the experiment within the first 24 hr after birth, with a mean ± standard deviation of 17.8 ± 0.4 hr for the 120 valid data.

Prior to data collection, written consent was obtained from the parents or legal guardians of all participating neonates for access to clinical information and EEG data collection for scientific purposes. While sample sizes were not statistically predetermined, including twenty subjects per GA group represented the maximum feasible number within a two-year period at Peking University First Hospital.

All subjects met the following inclusion criteria: (1) normal birth weight for their GA; (2) absence of clinical symptoms at the time of EEG recording; (3) no previous sedation or medication prior to EEG recording; and (4) normal hearing results in an evoked otoacoustic emissions test (ILO88 Dpi, Otodynamics Ltd, Hatfield, UK). Additionally, subjects did not exhibit any of the following neurological or metabolic disorders: (1) hypoxic-ischemic encephalopathy, (2) intraventricular hemorrhage or white matter damage detected by cranial ultrasound, (3) congenital malformation, (4) central nervous system infection, (5) metabolic disorder, (6) clinical evidence of seizures, and (7) signs of asphyxia.

### Stimuli
A total of 85 possible combinations of consonants and vowels, which are standard in Chinese (*Lee and Zee, 2003*) and common to most human languages (e.g. 'dada' and 'keke'), were recorded by a native Chinese-speaking adult woman with the Peking dialect. Each disyllable was recorded with four repetitions, two using a happy prosody and two with a neutral prosody, resulting in a total of 340 disyllables (85×4). Twenty Chinese undergraduate students (10 men, mean age 20.1 ± 1.2 years) performed a discrimination task, distinguishing between happy and neutral stimuli, and rated the affective content of these stimuli.

In the affective rating task, participants assessed the intensity of happiness (on a 9-point scale ranging from 1 being the least happy to 9 being the happiest) and the valence (on a 9-point scale ranging from 1 being the most negative, 5 being neutral, to 9 being the most positive) of the 340 stimuli. This study selected five pairs of happy and neutral disyllables that shared the same consonant-monophthong combinations and achieved 100% discrimination accuracy in the discrimination task (i.e. 'dada,' 'dudu,' 'gege,' 'keke,' and 'tutu' in Chinese Pinyin). Paired-samples t-tests demonstrated that the happy disyllables were rated as significantly happier ($t(4) = 24.70$, p<0.001; happy intensity: 7.49 ± 0.20 vs 3.53 ± 0.19) and had a more positive valence ($t(4) = 18.55$, p<0.001; valence: 7.11 ± 0.12 vs 4.91 ± 0.24) than their neutral counterparts. These ten disyllables were then standardized

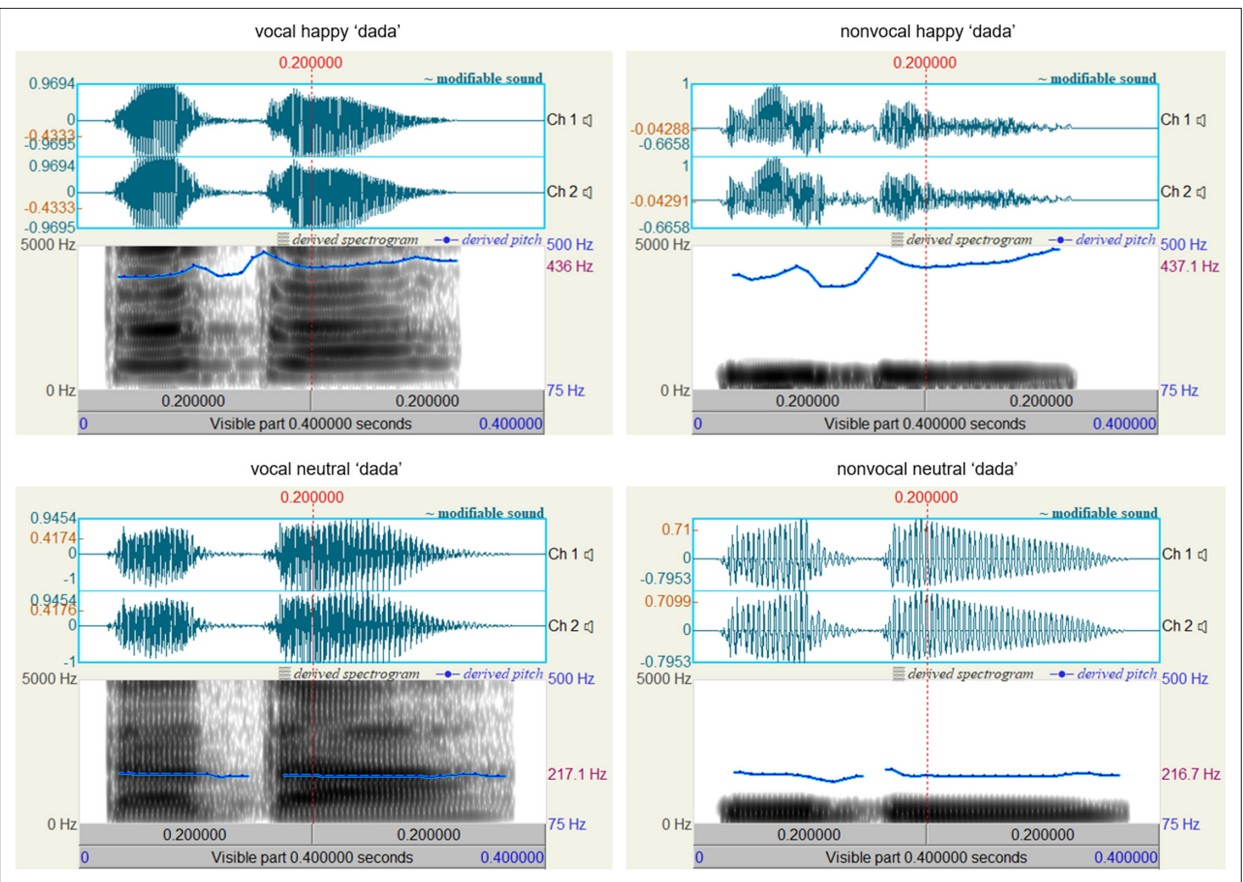

**Figure 3.** Oscillograms and spectrograms displaying vocal and nonvocal auditory stimuli for the syllable 'dada' (international phonetic symbol: [ta ta]).

to have the same mean intensity and had duration of 400ms using Adobe Audition (v.2022; Adobe Systems Inc, San Jose, CA).

To ensure that neonates were discriminating based on prosodic cues containing emotional content rather than low-level acoustic properties, we employed a method similar to *Cheng et al., 2012* and created a separate set of nonvocal control sounds. We hypothesized that the fundamental frequency (f0) alone does not convey emotional content in voices and that neonates require multiple other prosodic cues embedded in the high-frequent component to discern emotions, as suggested by *Cheng et al., 2012* and *Zhang et al., 2014*. As a result, ten nonvocal sounds were generated to match the f0 contours and temporal envelopes of their corresponding vocal sounds. This matching process was carried out using Matlab (v.2021b; MathWorks, Inc, South Natick, MA). Specifically, we initially applied a zero-phase filter with a bandpass of mean f0 ± 150 Hz to obtain f0-matched sounds of prosodic voices. Subsequently, a normalization procedure was implemented to ensure that the intensity of each pair of vocal and nonvocal sounds was equal. Oscillograms and spectrograms of the auditory stimuli utilized in this study are presented in *Figure 3*, generated using Praat (v.6.3.17, https://www.praat.org). All auditory stimuli, along with their pronunciations and rating scores, are available in the supplemental material labeled 'experimental sounds'.

To optimize the diversity of our material and increase the generalizability of our results, we utilized ten sets of sounds. Each set included both positive and neutral prosodic voices, along with their respective nonvocal counterparts. These auditory materials were distributed randomly and evenly within each neonatal GA group, ensuring that each set was presented twice (to two individuals) in each GA group.

## Procedure

The sound stimuli were presented in two blocks: the vocal and nonvocal conditions, utilizing the odd-ball paradigm. The standard stimulus was either a vocal or nonvocal neutral sound, while the

deviant stimulus was either a vocal or nonvocal happy sound. Each block consisted of 240 standard stimuli (80%) and 60 deviant stimuli (20%). The standard and deviant stimuli were presented randomly, ensuring that each deviant stimulus was followed by at least two standard stimuli. Each sound had a duration of 400 ms, and the inter-trial interval was silent, with varying durations ranging from 500 to 700 ms. Each block lasted for 5 min, and the order of the vocal and nonvocal blocks was counterbalanced across participants. A 5 min break separated the two blocks, resulting in a total EEG recording duration of 15 min.

The experiment took place in the neonatal ward of Peking University First Hospital. Neonates were transported to a designated testing room for EEG recording as soon as their condition stabilized after birth. In this room, they were separated from their mothers to minimize any natural exposure to speech or speech stimuli other than those utilized in the experiment. Auditory stimuli were presented through a pair of loudspeakers positioned approximately 30 cm away from the neonates' left and right ears, at a sound pressure level of 55–60 dB, with an average background noise intensity level of 30 dB. EEG recording was conducted while the neonates were in a natural sleep state (*Cheour et al., 2002*; *Wu et al., 2022*).

## Data recording and analysis

We recorded brain electrical activity using an electrical amplifier (NeuSen.W32, Neuracle, Changzhou, China) at a sampling frequency of 1000 Hz. Initially, the data were recorded online with reference to the left mastoid and subsequently re-referenced offline to the average of the left and right mastoids. The ground electrode was positioned on the forehead. For the recording of vertical eye movements, an electrooculogram (EOG) electrode was positioned beneath the left eye, while another was placed at the left external canthi for recording horizontal eye movements. Throughout the recording process, electrode impedances were meticulously maintained below 10 kΩ.

We eliminated ocular artifacts from the EEG data using a regression procedure implemented in NeuroScan software (Scan 4.3, NeuroScan, Herndon, VA). Subsequently, we employed Matlab (v.2021b; Mathworks, Inc, Sherborn, MA) for data processing and result presentation. The EOG-corrected EEG data were then offline filtered with a half-amplitude cutoff range of 0.01–30 Hz and segmented from 200 ms before sound presentation until 1000 ms after sound onset. Epochs were baseline-corrected relative to the mean voltage during the 200 ms preceding sound presentation. Any epochs containing artifacts with peak deflections exceeding ± 200 μV were rejected (see also *Biro et al., 2021*; *Di Lorenzo et al., 2021*; *Kumaravel et al., 2022*), followed by averaging for each experimental condition. The time window for the MMR component was pre-defined as 150–400 ms after sound onset, based on prior knowledge (*Cheour et al., 2002*), and utilized throughout the data analysis.

The number of valid epochs did not exhibit a significant difference across neonatal groups. In particular, for happy trials, a six (group) × two (condition: vocal/nonvocal) ANOVA was performed. Neither the main effect of condition ($F(1,114) = 0.533$, p=0.467, $\eta_p^2 = 0.005$) nor group ($F(5,114) = 0.795$, p=0.555, $\eta_p^2 = 0.034$) was found to be significant. The interaction between group and condition was also not significant ($F(5,114) = 1.654$, p=0.151, $\eta_p^2 = 0.068$). For neutral trials, another 6×2 ANOVA was performed. Again, neither the main effect of condition ($F(1,114) = 1.137$, p=0.289, $\eta_p^2 = 0.010$) nor group ($F(5,114) = 1.225$, p=0.302, $\eta_p^2 = 0.051$) was significant. The interaction between group

**Table 1.** Epoch numbers in different conditions (mean ± standard deviation).

| GA group (week) | Vocal happy | Nonvocal happy | Vocal neutral | Nonvocal neutral |
|---|---|---|---|---|
| 35 | 53.15 ± 4.86 | 52.65 ± 6.79 | 215.55 ± 16.58 | 214.10 ± 21.80 |
| 36 | 51.20 ± 7.18 | 48.80 ± 11.04 | 201.90 ± 30.78 | 196.25 ± 38.24 |
| 37 | 48.45 ± 6.79 | 50.40 ± 5.98 | 191.40 ± 38.00 | 199.70 ± 32.30 |
| 38 | 48.35 ± 9.03 | 51.00 ± 6.47 | 197.95 ± 35.37 | 202.60 ± 27.40 |
| 39 | 49.40 ± 7.06 | 49.65 ± 8.11 | 198.35 ± 31.65 | 198.25 ± 36.47 |
| 40 | 49.80 ± 6.53 | 50.35 ± 5.71 | 192.95 ± 32.55 | 202.85 ± 24.42 |

and condition was not significant (F(5,114) = 1.012, p=0.414, $\eta_p^2$ = 0.043). Epoch numbers in different conditions are reported in *Table 1*.

We performed statistical analyses using SPSS Statistics (v. 20.0; IBM, Somers, USA). Descriptive data are reported as mean ± standard deviation. The significance level was set at 0.05. We applied the Greenhouse-Geisser correction for ANOVA tests when deemed appropriate. Post-hoc tests for significant main effects were conducted using the Bonferroni method. Significant interactions were explored through simple effects models. We reported partial eta-squared ($\eta_p^2$) as a measure of effect size in ANOVA tests.

## Acknowledgements

This study was funded by the National High-Level Hospital Clinical Research Funding (High-Quality Clinical Research Project of Peking University First Hospital, 2022CR68), the National Natural Science Foundation of China (32271102; 31920103009), the Major Project of National Social Science Foundation (20&ZD153), Shenzhen-Hong Kong Institute of Brain Science (2024SHIBS0004), and the National Key Research and Development Program of China (2021YFC2700700). Funding statement. This study was funded by the National High Level Hospital Clinical Research Funding (High Quality Clinical Research Project of Peking University First Hospital, 2022CR68), the National Natural Science Foundation of China (32271102; 31920103009), the Major Project of National Social Science Foundation (20&ZD153), Shenzhen-Hong Kong Institute of Brain Science (2024SHIBS0004), and the National Key Research and Development Program of China (2021YFC2700700).

## Additional information

### Funding

| Funder | Grant reference number | Author |
|---|---|---|
| National High Level Hospital Clinical Research Funding | 2022CR68 | Xinlin Hou |
| National Natural Science Foundation of China | 32271102 | Dandan Zhang |
| National Natural Science Foundation of China | 31920103009 | Dandan Zhang |
| Major Project of National Social Science Foundation | 20&ZD153 | Dandan Zhang |
| Shenzhen-Hong Kong Institute of Brain Science | 2024SHIBS0004 | Dandan Zhang |
| National Key Research and Development Program of China | 2021YFC2700700 | Dandan Zhang |

The funders had no role in study design, data collection and interpretation, or the decision to submit the work for publication.

### Author contributions

Xinlin Hou, Conceptualization, Investigation, Writing – original draft; Peng Zhang, Data curation, Formal analysis, Investigation, Writing – original draft, Project administration; Licheng Mo, Data curation, Methodology, Writing – review and editing; Cheng Peng, Data curation, Writing – review and editing; Dandan Zhang, Conceptualization, Data curation, Formal analysis, Supervision, Funding acquisition, Visualization, Methodology, Writing – original draft, Project administration, Writing – review and editing

### Author ORCIDs

Dandan Zhang https://orcid.org/0000-0003-1825-7114

## Ethics

The research received approval from both the Ethical Committee of Peking University First Hospital and the Chinese Clinical Trial Registry (ChiCTR2300069898). Prior to data collection, written consent was obtained from the parents or legal guardians of all participating neonates for access to clinical information and EEG data collection for scientific purposes.

Reviewer #1 (Public Review): https://doi.org/10.7554/eLife.95393.4.sa1
Reviewer #2 (Public Review): https://doi.org/10.7554/eLife.95393.4.sa2
Author response https://doi.org/10.7554/eLife.95393.4.sa3

---

# Additional files

## Supplementary files

• Supplementary file 1. Experimental sounds.
• Source code 1. Contains the MATLAB script.

## Data availability

The experimental materials are available as supplementary files for download. EEG epochs from all 120 datasets can be accessed at https://doi.org/10.17605/OSF.IO/A3XZY. A Matlab script detailing the preprocessing steps and how to extract the EEG epochs from the raw data is also provided as a *Source code 1*. The data and materials shared in this study are available for academic purpose free of charge, provided that proper citation of this article is given. The raw EEG data have not been disclosed because the parents of 89 out of the 120 neonates did not consent to making their babies' raw neurobiological data publicly available. Noncommercial researchers interested in accessing the original raw EEG data must submit a project proposal. This proposal will be reviewed by the Ethical Committee of Peking University First Hospital to determine whether access can be granted to the 31 datasets for which parents permitted sharing of the raw data solely for academic purposes. For further inquiries, please contact the corresponding author.

The following dataset was generated:

| Author(s) | Year | Dataset title | Dataset URL | Database and Identifier |
|---|---|---|---|---|
| Li Y | 2024 | newborn EEG | https://doi.org/10.17605/OSF.IO/A3XZY | Open Science Framework, 10.17605/OSF.IO/A3XZY |

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
