## [Editor Report · eLife assessment]

This is an **important** study on changes in newborns' neural abilities to distinguish auditory signals at 37 weeks of gestation. The evidence of change in neural discrimination as a function of gestational age is **convincing**, but, as the authors acknowledge, further control of the acoustic signals and infants' language environment is necessary for the results to be used in clinical applications. The work contributes to the field of neurodevelopment.

---

## [Referee Report · Reviewer #1 (Public Review)]

Summary:

This manuscript aimed to investigate the emergence of emotional sensitivity and its relationship with gestational age. Using an oddball paradigm and event-related potentials, the authors conducted an experiment in 120 healthy neonates with a gestational age range of 35 to 40 weeks. A significant developmental milestone was identified at 37 weeks gestational age, marking a crucial juncture in neonatal emotional responsiveness.

Strengths:

This study has several strengths, by providing profound insights into the early development of social-emotional functioning and unveiling the role of gestational age in shaping neonatal perceptual abilities. The methodology of this study demonstrates rigor and well-controlled experimental design, particularly involving matched control sounds, which enhances the reliability of the research. Their findings not only contribute to the field of neurodevelopment, but also showcase potential clinical applications, especially in the context of autism screening and early intervention for neurodevelopmental disorders.

---

## [Referee Report · Reviewer #2 (Public Review)]

This is an important and very interesting report on a change in newborns' neural abilities to distinguish auditory signals as a function of the gestational age (GA) of the infant at birth (from 35 weeks GA to 40 weeks GA). The authors tested neural discrimination of sounds that were labeled 'happy' vs 'neutral' by listeners that represent two categories of sound, either human voices or auditory signals that mimic only certain properties of the human vocal signals. The finding is that a change occurs in neural discrimination of the happy and neutral auditory signals for infants born at or after 37 weeks of gestation, and not prior (at 35 or 36 weeks of gestation), and only for discrimination of the human vocal signals; no change occurs in discrimination of the nonhuman signals over the 35- to 40-week gestational ages tested. The neural evidence of discrimination of the vocal happy-neutral distinction and the absence of the discrimination of the control signals is convincing. The authors interpret this as a 'landmark' in infants' ability to detect changes in emotional vocal signals, and remark on the potential value of the test as a marker of the infants' interest in emotional signals, underscoring the fact that children at risk for autism spectrum disorder may not show the discrimination.

[Editors' note: The authors addressed the reviewers' main concern by clearly stating the limits of the study's implications.]

---

## [Author Response]

The following is the authors’ response to the previous reviews.

**Reviewer #1:**
After reviewing the authors' response letter and the revised manuscript, I believe they have done a commendable job in addressing my comments.Additionally, I concur with the concerns raised by Reviewer #2 regarding several potential confounding factors that require better control in their experimental design. These include the differences in physical properties between vocal and nonvocal stimuli, as well as the infant's exposure to the speech/auditory environment. These concerns should be thoroughly and explicitly discussed in the manuscript, ensuring a clearer understanding for the readers.

Thank you for the suggestion. We have discussion these limitations in our revised manuscript. In this round of revision, we have tempered our conclusion due to these limitations.

**Reviewer #2:**
The revised manuscript does discuss the limitations of the control stimuli, as well as the limitations with regard to conclusions that can be drawn from this data set. I therefore expected the authors to temper a bit their recommendation that this could be a 'screening' signal for autism because these data are not sufficiently strong to make that recommendation. Also, in the same vein, perhaps the title might be adjusted somewhat to suggest less certainty, for example, by using the word "change" rather than "milestone"'? The data are of interest, but the limitations are genuine limitations.

Thank you for your expert comments and considerations. We have moderated our recommendation for autism screening and softened the statement of “milestone” throughout the manuscript. Please see the updated article title, abstract, significance statement, and discussion.